# Harderian Gland Development and Degeneration in the *Fgf10*-Deficient Heterozygous Mouse

**DOI:** 10.3390/jdb12020016

**Published:** 2024-06-03

**Authors:** Shiori Ikeda, Keita Sato, Hirofumi Fujita, Hitomi Ono-Minagi, Satoru Miyaishi, Tsutomu Nohno, Hideyo Ohuchi

**Affiliations:** 1Department of Cytology and Histology, Medical School, Okayama University, 2-5-1 Shikata-cho, Kita-ku, Okayama 700-8558, Japan; 2Department of Cytology and Histology, Faculty of Medicine, Dentistry and Pharmaceutical Sciences, Okayama University, 2-5-1 Shikata-cho, Kita-ku, Okayama 700-8558, Japan; 3Department of Cytology and Histology, Graduate School of Medicine, Dentistry and Pharmaceutical Sciences, Okayama University, 2-5-1 Shikata-cho, Kita-ku, Okayama 700-8558, Japan; 4Department of Legal Medicine, Faculty of Medicine, Dentistry and Pharmaceutical Sciences, Okayama University, 2-5-1 Shikata-cho, Kita-ku, Okayama 700-8558, Japan

**Keywords:** Harderian gland, *Fgf10*, haploinsufficiency, mouse

## Abstract

The mouse Harderian gland (HG) is a secretory gland that covers the posterior portion of the eyeball, opening at the base of the nictitating membrane. The HG serves to protect the eye surface from infection with its secretions. Mice open their eyelids at about 2 weeks of age, and the development of the HG primordium mechanically opens the eye by pushing the eyeball from its rear. Therefore, when HG formation is disturbed, the eye exhibits enophthalmos (the slit-eye phenotype), and a line of *Fgf10^+/−^* heterozygous loss-of-function mice exhibits slit-eye due to the HG atrophy. However, it has not been clarified how and when HGs degenerate and atrophy in *Fgf10^+/−^* mice. In this study, we observed the HGs in embryonic (E13.5 to E19), postnatal (P0.5 to P18) and 74-week-old *Fgf10^+/−^* mice. We found that more than half of the *Fgf10^+/−^* mice had markedly degenerated HGs, often unilaterally. The degenerated HG tissue had a melanized appearance and was replaced by connective tissue, which was observed by P10. The development of HGs was delayed or disrupted in the similar proportion of *Fgf10^+/−^* embryos, as revealed via histology and the loss of HG-marker expression. In situ hybridization showed *Fgf10* expression was observed in the Harderian mesenchyme in wild-type as well as in the HG-lacking heterozygote at E19. These results show that the *Fgf10* haploinsufficiency causes delayed or defective HG development, often unilaterally from the unexpectedly early neonatal period.

## 1. Introduction

The Harderian gland (HG) is an orbital gland attached to the back of the eyeball, which most tetrapods, terrestrial animals, and most of primates, except for humans, have [1]. The HG was first described by a Swiss physician and anatomist, Johann Jacob Harder, in 1694 [2]. Functions of HGs have been explored and include a lubricant in the orbit, the production of pheromones as part of the vomeronasal system, extraretinal photoreception, water repulsion and the thermoregulation of the fur, a local immune response at the upper airway, etc. However, its common roles conserved beyond species have been elusive [3]. In humans, primordia for the HG are degenerated at 23 to 30 weeks of gestation [4].

The mouse HG appears to be speckled with dark-brown pigments due to porphyrin in the tubuloalveolar lumens and is dark grey due to melanocytes in the stroma [5,6,7]. There is no lobulation, no branched duct, and the lumens are wide. The tubuloalveolar gland consists of columnar epithelial cells, whose cytoplasm is eosinophilic with many clear secretory granules containing lipids. Other than the more numerous type A cells for secreting lipid droplets, there are type B cells for producing multilamellar vesicles [8,9], which are surrounded by basal myoepithelial cells.

Although Harder first described that the HG is “a new lacrimal gland” (LG), the LG develops from the temporal epithelium of the eye, while the HG arises from its nasal epithelium. In contrast to the HG, the LG has distinct lobules, branched ducts, and alveoli with narrow lumens [9]. To protect the eye surface, most animals have a third eyelid, the nictitating membrane, which moves horizontally. The duct of the HG opens at the base of the nictitating membrane, but as mentioned, both the nictitating membrane and the HG have dwindled away in humans, and the former has become a small structure known as the plica semilunaris. Since in the model animal, the mouse, the HG is a conspicuously large retrobulbar exocrine gland, while in humans it is totally lacking, it would be intriguing to know the evolutional significance of the gain/loss of the HG. 

Regarding the genes involved in the formation of the HG, there are several mutant mice deficient in HG development. In the *Fibroblast growth factor 10* (*Fgf10*)-null mouse, the HG does not develop at all [10,11], indicating that *Fgf10* is required for the formation of HGs. Since the heterozygous knockout mutants for *Fgf10* in mice and humans were found in the deficiency of the lacrimal and salivary glands [12], one allelic expression of *Fgf10* was insufficient for the complete formation of the secretory glands. As for mice deficient in transcription factor genes, *Barx2*-null mice show the absence of HGs, while *Sox10*-conditional knockout mice have no secretory HG acini at postnatal day three [13,14]. These genes encode transcription factors that cooperate with *Fgf10* or are activated by FGF-signaling. In the protein phosphatase 2A (*pp2a*) (L309A) transgenic mice in which glandular cell adhesion is reduced, HG formation is disrupted in early postnatal days, and the glands become atrophied and replaced with connective tissues by postnatal day 18 [15]. Pp2a is also known as a downstream of FGF-signaling to inhibit ERK [16]. Thus, FGF-signaling and its downstream are crucial for HG formation. 

Puk et al. reported that another *Fgf10*-heterozygous knockout (*Fgf10^+/−^*) mouse line, *Aey17*, whose point mutation was made by N-ethyl-N-nitrosourea mutagenesis, was atrophying in HGs, leading to corneal disorders and other anterior ocular infections [17]. Here, we examine the HG phenotypes in the *Fgf10^+/−^* mouse that we had, where the ATG-containing exon was disrupted by the insertion of a neomycin gene cassette [18] to know when and how the HG was affected under the *Fgf10* haploinsufficiency. 

## 2. Materials and Methods

### 2.1. Genotyping of Fgf10-Mutant Mice

DNA was extracted from the tail tips of offspring and genotyped by PCR using TaKaRa Ex Taq DNA polymerase (Takara Bio, Kusatsu, Japan) and the four specific primers: P1, 5′-CTTCCAGTATGTTCCTTCTGATGAGAC-3′; P2, 5′-GTACGGACAGTCTTCATTCTTGGTCCC-3′; P3, 5′-ACGACGGGCGTTCC TTGCGCAGCTGTG-3′; P4, 5′-TCAGAAGAACTCGTCAAGAAGGCGATA-3′ [18]. The animal study protocol was approved by the Institutional Review Board of Okayama University (protocol codes OKU-2020404, OKU-2023243) for studies involving animals.

### 2.2. Fixation, Sectioning, and Histology

After transcardiac perfusion with 4%PFA, Harderian glands were dissected out and fixed in the same fixative or Fekete’s solution (10 mL of 37% formaldehyde, 100 mL of 70% ethanol, 5 mL of glacial acetic acid) [19] for 24 h. After being washed in PBS or in 70% ethanol, the samples were dehydrated in ethanol, cleared in xylene, and embedded in paraffin. For histological analysis, paraffin sections (5 μm) were prepared, and deparaffinized sections were stained with hematoxylin (Mayer’s hemalum solution; Merck #1.09249.0500; Merck, Darmstadt, Germany) and eosin (Eosin Y-solution 0.5% aqueous; Merck #1.09844.1000) according to standard procedures. To reveal connective tissues, Masson trichrome staining was performed. Deparaffinized sections were also used in the immunostaining of pancytokeratin and TUNEL staining. 

### 2.3. Immunohistochemistry

For phalloidin staining, frozen sections (15 μm) were prepared (on a cryostat, Tissue-Tek Polar-B; Sakura Finetek Japan, Tokyo, Japan) from cryoprotected (with 20% sucrose in PBS) samples embedded in an O.C.T. compound (Tissue-Tek 4583, Sakura Finetek). For immunohistochemistry, an ImmPRESS kit (Vector Laboratories, Newark, CA, USA) was used and visualized with alkaline phosphatase and a Vector Red substrate (ImmPACT Vector Red, Vector Laboratories). Nuclei were stained with DAPI or Hoechst. The antibodies and other reagents used in this study are listed in Appendix A. As negative controls, normal IgG was used instead of primary antibodies. 

### 2.4. TUNEL Staining 

To detect dying cells, TUNEL staining was performed using a kit, the DeadEnd™ Colorimetric TUNEL System (Promega, Madison, WI, USA) or DeadEnd™ Fluorometric TUNEL System (Promega). The Proteinase K treatment for the E19 head (15 μm frozen sections) and the P18 HG (5 μm deparaffinized sections) was 10 μg/mL at room temperature for 10 min. On negative control sections, terminal deoxytransferase (TdT) was omitted. 

### 2.5. In Situ Hybridization (ISH)

Conventional ISH was performed on frozen sections (15 μm) using digoxigenin-labelled riboprobes as previously described [20]. The cDNAs corresponding to the 5′-untranslated and coding regions for *Fgf10* and *Fgf7,* and the coding region of *Fgfr2b,* were amplified from mouse head cDNA and inserted into the pBluescript KS vector. Riboprobes were prepared with DIG RNA labeling mix (Roche, Basel, Switzerland). For *Fgf10* and *Fgf7*, antisense and sense riboprobes were transcribed with SP6 RNA polymerase and template DNA were amplified using gene-specific primers flanked with SP6 promotors at their 5′ end. Antisense and sense riboprobes of *Fgfr2b* were transcribed with T7 and T3 RNA polymerases, respectively, and template DNA were amplified using T7 and T3 primers. The primer sequences to generate template DNAs for the riboprobes are listed in Appendix A. For in situ hybridization, tissue sections were treated with 0.5 μg/mL Proteinase K for 15 min at room temperature. Riboprobes were applied at 0.17 μg/mL and were hybridized at 65 degrees for two days. After reaction with alkaline phosphatase (AP)-conjugated anti-DIG antibody, color development was performed using AP substrate NBT/BCIP at 28 degrees for two days. SABER-FISH for dual ISH to detect *Fgf10* and *Fgfr2b* transcripts simultaneously was performed according to Kishi et al. [21], with slight modifications, as previously described [22]. DNA oligos used in SABER-FISH are listed in Appendix A. 

### 2.6. Microscopy and Camera/Imaging System

Eyes and HGs were photographed with a digital camera system (Leica DFC310; Leica Microsystems, Wetzlar, Germany). Tissue sections were observed with bright-field or differential interference contrast microscopy (Leica DM5000B) and photographed with a CCD camera system (Nikon DS-Fi1; Nikon, Tokyo, Japan). Brightfield images of in situ hybridization on tissue sections were also captured with an IDS UI-3290SE-C-HQ camera (Imaging Development Systems, Obersulm, Germany) mounted on a Carl Zeiss Axioplan microscope (Carl Zeiss, Oberkochen, Germany). Fluorescent micrographs (Figure 1P,Q and Appendix A) were taken with a Leica upright DM5000B microscope equipped with a Nikon DS-Qi1 CCD camera system. Confocal fluorescence images were collected with a Carl Zeiss LSM 780 laser scanning confocal microscope system with 405, 488, and 561 nm laser lines. Brightness and contrast adjustments were performed for some images, and image manipulation was performed using a ZEN 2012 SP1 black edition.

### 2.7. Statistics

Significance was determined using a Kruskal–Wallis test and a Steel-Dwass test (Figure 1M). Data were presented in box and whisker plots as upper and lower extremes, upper and lower quartiles, medians, and outliers. Values of *p* < 0.05 were considered significant. Data analysis was performed using RStudio (https://posit.co/; accessed on 29 November 2023) with R version 4.2.2. 

## 3. Results

### 3.1. More Than Half the Cases of Fgf10^+/−^ HGs Exhibiting, often Unilaterally, the Loss of Glandular Cells, Fibrosis and Melanocytic Hyperplasia as Early as Postnatal Day Six

We first observed HG phenotypes in wild-type (WT) and *Fgf10^+/−^* mice at 74 weeks of age, the oldest mice obtained. Since there is sexual dimorphism in hamster and mouse HGs [23,24,25] and in mouse salivary glands including lacrimal glands [26,27,28], we mainly focused on HGs from male, immature, prepuberty mice in this study. At approximately one and a half years of age, the unilateral atrophy of HGs was observed in more than half of male *Fgf10^+/−^* mice, but not in WT HGs (Appendix A and Figure 1A–D). The atrophied HG was markedly small and blackened (Figure 1C,C’). Histology showed that the WT eyeball had the HG lobes of well-developed tubuloalveolar gland cells in the rear (Figure 1E–G and Appendix A). In contrast, the affected *Fgf10^+/−^* HG exhibited fibrosis and was partly replaced with adipose tissue in the Harderian capsule (Figure 1H, arrow; Figure 1I; compare with the contralateral side, Figure 1J and Appendix A).
Figure 1The Harderian gland phenotype of the *Fgf10* heterozygous knockout mouse: the unilateral gland is often degenerated within 2 weeks after birth. (**A**–**D**) A lateral view of the right (**A**,**C**,**C**’) or left (**B**,**D**) eyeball with a Harderian gland (HG, arrows) in the rear, from wild-type (WT) (**A**,**B**) and an *Fgf10^+/−^* mice (**C**,**C’**,**D**) at 74 weeks. Panel (**C’**) is a high magnification of (**C**). (**E**–**J**) The histology of a WT or an *Fgf10^+/−^* eye with a HG (arrows) of 74-week-old mice. Hematoxylin-eosin (HE) (**E**,**H**) or Masson trichrome (MT) (**F**,**G**,**I**,**J**) staining was performed. The latter stain reveals connective tissue and collagen fibers in blue. Left (**E**,**F**,**J**) and right (**G**,**H**,**I**) sides are shown. The section in (**H**) shows a relatively end portion of the lens to demonstrate the degenerated HG (arrow **I**). (**K**,**L**) Isolated HGs at postnatal day 18 (P18). The HG on the left side (**L**) is small and blackened, compared with the HG on the right side (**K**). (**M**) A comparison of the HG weight (mg) per body weight (gram) between WT HGs (blue bar), normal HGs from *Fgf10^+/−^* mice (*Fgf10^+/−^* N; orange bar), and small HGs from *Fgf10^+/−^* (*Fgf10^+/−^* S; gray bar) mice. *p*-value = 0.0007899 in the Kruskal–Wallis test; *p*-values = 0.0015 (*Fgf10^+/−^* normal vs. *Fgf10^+/−^* small), 0.4943 (WT vs. *Fgf10^+/−^* normal), and 0.002 (WT vs. *Fgf10^+/−^* small) by the Steel-Dwass test. Source data are included in Appendix A. (**N**,**O**) Masson trichrome staining of *Fgf10^+/−^* (40F_#7) right and left HG sections, respectively. (**P**,**Q**) TUNEL (**P**) (green) and (**Q**) nuclear (blue) stains of right HG sections from an *Fgf10^+/−^* mouse at P18 (40F_#7). (**R**–**U**) Immunohistochemistry of pancytokeratin (red). Sections of HGs on both sides from a WT (**R**,**S**) and an *Fgf10^+/−^* (**T**,**U**) mice were used. Interstitial melanocytes are observed in dark brown. Fibrosis and melanocytic hyperplasia are observed in (**I**,**O**,**U**). (**V**–**Y**) Histology of *Fgf10^+/−^* HGs at P6. Consecutive sections through upper to lower horizontal planes were observed and representative sections are shown. In the #2 mouse, HG cells are absent on the right side (arrowheads in **V**), while they have developed on the left side (**W**). In the #3 mouse, HG cells are shrunk and degenerating on the right side (arrow in **X**), while they have developed on the left side (**Y**). The portion containing degenerating glandular epithelial cells (arrow in **X**) is enlarged in (**X’**). (**Z**–**AA’**) The TUNEL staining of *Fgf10^+/−^* head sections at P6 (**Z**–**Z”** from #2, **AA** and **AA’** from #3). Arrowheads in (**Z**) indicate the Harderian capsule. The boxed area in (**Z**) contains two TUNEL-positive portions (green). The left is enlarged in (**Z’**) and the right is enlarged in (**Z”**). (**AA’**) The 3D structure of the TUNEL-positive portion in (**AA**, green). The section of (**AA**) is approximately 60 μm caudal to the section of (**X**), suggesting that the TUNEL-positive cell in (**AA**) is part of the degenerating glandular cell population in (**X’**). Abbreviations: el, eyelid; em or m, extraocular muscle; hg, Harderian gland; on, optic nerve; re, retina. Scale bars: 1 mm (**A**–**D**,**K**,**L**), 400 μm (**E**,**H**,**G**,**J**), 200 μm (**F**,**I**,**V**–**Y**), 100 μm (**N**,**O**,**P**,**Q**,**R**–**U**,**X’**,**AA**), 50 μm (**Z**), and 5 μm (**Z’**,**Z”**).
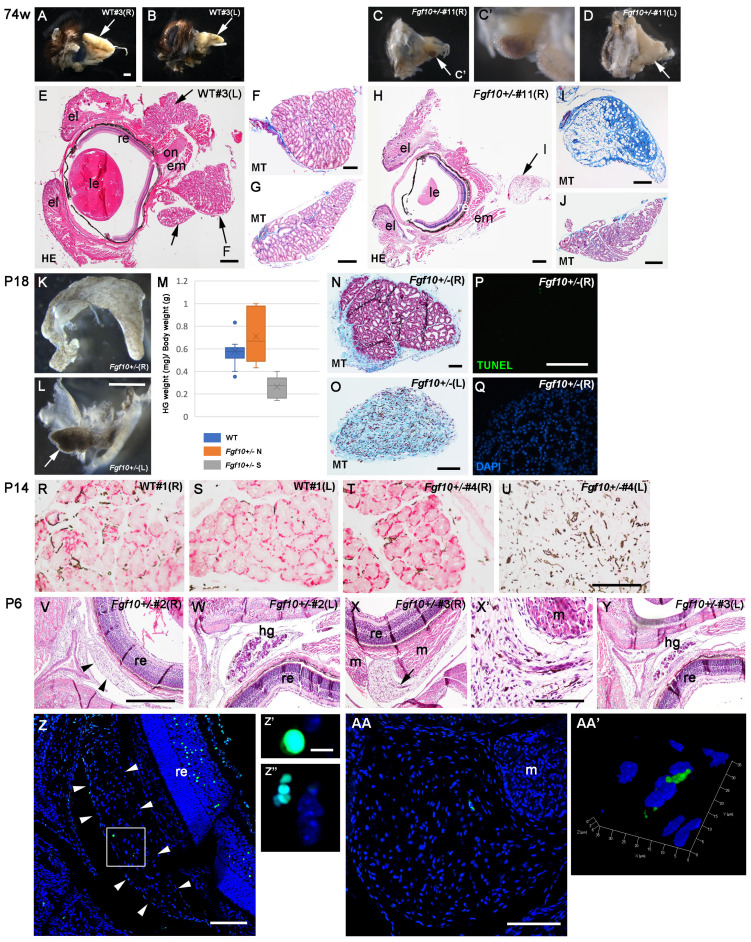


We next examined when the HG became affected in the *Fgf10^+/−^* mice. Mice open the eyelid at about 2 weeks postnatal [29] and we also observed the slit-eye phenotype, enophthalmos in our *Fgf10^+/−^* mice, as reported in [17] (Appendix A). Therefore, we explored HGs at postnatal day 18 (P18) after complete eye-opening at a norm (Appendix A, Figure 1K,L). Dissecting off the HG was followed by measuring its weight to normalize with the body’s weight. There were already affected HGs, which were small and blackened in the *Fgf10^+/−^* mice (Figure 1L; compare with the contralateral side, Figure 1K). We compared the HG weight per body weight between three groups (Figure 1M): WT HGs, normal-looking *Fgf10^+/−^* HGs, and decreased *Fgf10^+/−^* HGs. The normalized HG weight was significantly light in the decreased *Fgf10^+/−^* HGs. Histology showed WT and normal-looking HGs in *Fgf10^+/−^* have tubuloalveolar gland cells and thin capsule and interlobular connective tissues (Figure 1N). In contrast, the decreased *Fgf10^+/−^* HG was already replaced by fibrous tissues with melanocytic hyperplasia (Figure 1O; four out of six decreased HGs in *Fgf10^+/−^*) or there were still glandular cells remained but fibrosis emerging (two out of six decreased HGs). TUNEL staining showed there was no marked cell death at P18 in normal-looking *Fgf10^+/−^*HGs (Figure 1P,Q) (n = 5), suggesting that these *Fgf10^+/−^* HGs would develop normally. At P14 (Appendix A), similar fibrosis and melanocytic hyperplasia in the place of glandular cells was often unilaterally observed in the *Fgf10^+/−^* HG (Appendix A, two out of three).

It was reported that the *Fgf10^+/−^* atrophied HGs had lost the expression of pancytokeratin [17]. Immunohistochemistry showed that WT HGs exhibited a localization of pancytokeratin at the base of the glandular cells at P14 (Figure 1R,S). In contrast, the *Fgf10^+/−^* atrophied HGs lost pancytokeratin (Figure 1U), while the contralateral normal-looking HG in the same heterozygote retained the expression (Figure 1T). There was also a heterozygote in which HGs on both sides retained pancytokeratin expression in the glandular cells (Appendix A; compare with a WT control, Appendix A). The histology of *Fgf10^+/−^* HGs at P10 showed the unilateral absence of HGs in the heterozygote, in which similar, fibrotic histological change was observed (Appendix A), while at P6, *Fgf10^+/−^* typical HG primordia were already absent (Figure 1V) or the glandular cells appeared to be shrunk and degenerating (Figure 1X,X’, arrow) on the right side. A few TUNEL-positive cells were detected in the degenerated (Figure 1V,Z–Z”, Appendix A) and degenerating (Figure 1X,X’,AA,AA’, Appendix A) HGs. These results indicate that *Fgf10^+/−^* HGs, in more than half the cases, had unilateral atrophies as early as P6.

### 3.2. More Than Half the Cases of Fgf10^+/−^ HGs Exhibit the Unilateral Absence of HG Development Just before or after Birth

The primordia of the lacrimal glands develop from the primitive conjunctival epithelium on the temporal side (Figure 2A,B, arrowheads), while those of HGs derive from the nasal primitive conjuntiva (Figure 2A,B, arrows) [10]. The duct of the HG opens at the base of the third eyelid, the nictitating membrane, which has a cartilage core. By embryonic day 15 (E15), the primordium for the nictitating membrane (nm) emerges via the further invagination of the primitive conjunctival epithelia flanking the developing nm (Figure 2C,C’; nm is the asterisk in Figure 2C’). These invaginating epithelia on the nasal side are likely primordia for HGs. The observation of serial histological sections revealed that the two invaginating epithelia flanking the nm exhibited asymmetric elongation: the tip of the invaginating epithelium along the eyeball was multilayered with columnal and cuboidal cells (Figure 2C’, Appendix A). We then focused on the embryonic stage to see when *Fgf10^+/−^* HG primordia would fail to develop or be delayed in their development. At E13.5, we could not identify obvious morphological changes in the emerging HGs between WT and *Fgf10^+/−^* embryos (Figure 2A,B’) (n = 2 for each genotype). At E15, the *Fgf10^+/−^* invaginating epithelium exhibited less development and was variable in elongation and expansion, compared with that of the wild-type (n = 3 for each genotype) (Figure 2D,D’, Appendix A). 

At E19, just before birth, histological examination shows sections of proliferating Harderian glandular cells with elongating and cavitating (Figure 2E). A proliferating cell marker, Ki67, was localized to the non-cavitating HG cell nuclei as well as those of developing retinal cells (Figure 1F), while its expression was downregulated in the cavitating glandular cells (Figure 1F, arrow; Appendix A). TUNEL staining shows cell death is observed in the cells abutting the lumen (Figure 2G, arrows; Figure 2H as a negative control), where pyknotic nuclei are detected by the HE stain. The HG consists of tubuloalveolar gland cells and has a wide lumen in acinus, characteristic of lipid-containing secretions from this gland [9]. Just like the cavitation of mammary glands mediated by apoptosis or autophagy [30,31], a similar mechanism may be involved in the tubulogenesis of the HG. We examined WT and *Fgf10*^+/−^ HGs just before and after birth (at E19 and P0.5): horizontal sections of the HE stain reveal that the developing HG cells are surrounded by sheath-like mesenchyme filled with fibroblasts and melanocytes (Figure 2E,I,J). The observation of consecutive sections through upper to lower horizontal planes can identify the presence or absence of glandular cells within the Harderian mesenchyme. In *Fgf10^+/−^* embryos, more than half the cases exhibit a unilateral absence of HG development (Figure 2L; compare with Figure 2I,J,K). As shown [17], pancytokeratin is a differentiation marker for mature HG cells, such as those at P14 (Figure 1R) and at 10 weeks postnatal (Appendix A). We first confirmed that pancytokeratin was already present in the developing HG cells at P0.5 (Appendix A). Then, we examined whether the developing *Fgf10^+/−^* HGs retained pancytokeratin protein (Figure 2I–P and Appendix A) (n = 2 for each genotype). We identified that in newborn mice that were lacking in HG development on one side (Figure 2L and Appendix A), pancytokeratin was absent on that side (Figure 2P and Appendix A). These results indicate that more than half the case of *Fgf10^+/−^* HG primordia fail to develop or are delayed in development unilaterally just around birth.

### 3.3. A Re-Examination of the Expression Pattern of Fgf10 in Association with Fgfr2b and Fgf7 during Mouse HG Development

Since the HG and lacrimal gland are absent from *Fgf10*-null mice, the expression pattern of *Fgf10* was examined to specify the role for FGF10 during ocular gland development [10]. Accordingly, it is now thought that *Fgf10* is not required for the initial development of these ocular glands, but for their proliferation and morphogenesis phase after E15 [10]. Here, we re-examined the expression pattern of *Fgf10* in the periocular mesenchyme, combined with those of *Fgfr2b,* a major FGF10-receptor gene as a marker for developing epithelia, and *Fgf7* as an *Fgf10*-paralogous gene [32]. At E15, relatively intense *Fgf10* mRNA signals were detected in an oval or crescent domain of the periocular mesenchyme, located nasolaterally (Figure 3A, black arrowhead), and in the mesenchyme that surrounds developing nasal glands (Figure 3A, red arrowhead), and nasal septum mesenchyme (Figure 3A, right lower corner of the panel). Lower levels of *Fgf10* transcripts were detected in the developing optic nerve, and in the mesenchyme near the extraocular muscles. In the three regions where *Fgf10* expression was relatively intense, that is, the oval periocular domain, the nasal gland mesenchyme, and the septum mesenchyme, an antisense strand of *Fgf10* was also transcribed (Figure 3B). In contrast, *Fgfr2b* was expressed in the developing epithelia consisting of future skin, eyelid, conjunctiva, cornea, HGs, superficial glands of the third membrane at relatively high levels, and nasal epithelia at much lower levels (Figure 3C). *Fgfr2b* was also expressed in the future orbit mesenchyme and nasal deep mesenchyme at relatively high levels, while it was expressed in the periocular mesenchyme where *Fgf10* was expressed at a relatively lower level (Figure 3C, arrowhead). As for *Fgf7*, diffuse expression was observed in the periocular mesenchyme (Figure 3E), compared with the result with a sense *Fgf7* probe (Figure 3F). 

We next focused on the developing HGs (the same as deep glands of the third eyelid) and superficial glands of the third eyelid under high magnification (Figure 3G–J). We found that *Fgf10* was expressed in the mesenchyme abutting the emerging superficial gland (Figure 3G, black arrow), which is away from the invaginating epithelium (red arrows in Figure 3G) that flanks the nascent nictitating membrane (nm) (Figure 3G, asterisk). Since the HG has a single duct, while those of superficial glands of the nm have many ducts [3], the upper invaginating epithelium seems to develop the HG duct and acinar cells with wide lumen (Figure 3L). As mentioned, *Fgfr2b* expression visualizes the future corneal epithelium, eyelid and nm epithelia, and the superficial glands (Figure 3H, black arrow). In contrast to the *Fgf10* expression domain, *Fgf7* was expressed in the mesenchyme abutting the invaginating nm epithelium (Figure 3I, red arrows), whose expression level seems to be relatively weak but distinct, compared with the result with the sense *Fgf7* probe (Figure 3J). 

At E19, *Fgf10* was expressed in the sheath-like Harderian mesenchyme (Figure 4A, arrowheads) along with being expressed in a deep mesenchymal layer of the developing eyelid (Figure 4A, red arrow; Figure 4B as a sense control). High magnification showed that *Fgf10* was not expressed by developing hemangioblasts (Appendix A, red arrowheads) at E15 or melanocytes in the Harderian mesenchyme at E19 (Appendix A as sense controls). *Fgfr2b* was not expressed in the hemangioblast at E15 either (Appendix A). *Fgf10* was not expressed in the mesenchyme near the elongating and cavitating HGs at E19 (Appendix A, arrow). These results indicate that *Fgf10* is expressed in the distinct periocular mesenchymal domains at E15 and E19 when the developing HGs are in the proliferation and morphogenesis phase. Regarding developing HGs, it is noteworthy that the distinct mesenchymal domain of *Fgf10* expression is not abutting the nascent HG epithelium, but is away from it.

### 3.4. Fgf10 Expression in the Harderian Mesenchyme Lacking Developing Glandular Cells Just before Birth

It was reported that the overexpression of *Fgf7* in the eye leads to ectopic gland formation, similar to HGs [33]. Given that *Fgf7* is a paralogous gene to *Fgf10,* with both activating the same receptor tyrosine kinase, FGFR2b, FGF7 might somehow contribute to HG formation when *Fgf10* expression is reduced. Therefore, we first examined whether *Fgf10* mRNA were reduced in *Fgf10^+/−^* heads at E17.5. Quantitative PCR showed that the relative expression level of *Fgf10* was significantly reduced in the *Fgf10^+/−^* embryonic heads, compared with wild-type heads (Appendix A). In contrast, the *Fgf7* expression level was not significantly altered in the *Fgf10^+/−^* heads compared to the wild-type heads (Appendix A). In situ hybridization (ISH) also showed that the *Fgf7* expression pattern was not altered in the *Fgf10^+/−^* developing Harderian tissues at E19, regardless of whether the glandular cells emerged or not, compared to that in the wild-type tissues (Figure 4C–H and Appendix A). 

*Fgf10* is a causal gene for human disorders such as the aplasia of the lacrimal and salivary glands (ALSG), in which one allele for *Fgf10* is disrupted and non-functional in patients [12]. This indicates that *Fgf10* is a haploinsufficiency gene: haploinsufficiency refers to a phenotype associated with the inactivation of a single allele in a diploid organism. As an attempt to clarify why the gene product from a residual single allele is not sufficient for developing a certain structure, a stochastic model of gene expression has been postulated [34]. Recently, stochastic gene expression, or transcriptional bursting, can be visualized in the transgenic cells of animals [35]. The activation and deactivation kinetic rate are dependent on the gene, and if the gene has a slow rate of gene expression kinetics, there will be time where there is a lack of a gene product required by the transcription of only one allele. We therefore examined whether ISH could detect the absence of *Fgf10* transcription in HG-lacking *Fgf10^+/−^* tissues. We found that *Fgf10* was expressed in the Harderian mesenchyme even in HG-lacking *Fgf10^+/−^* tissues (Figure 4I and Appendix A, arrowhead). *Fgfr2b* expression verified that the developing HGs were not contained in the Harderian mesenchyme examined (Figure 4K, arrowhead; Appendix A). These results suggest that ISH could not detect the stochastic absence of *Fgf10* transcription and that the loss of HG primordia or the delayed development of HGs was not dependent on the *Fgf10* expression at E19, but rather on earlier stages before E19. 

## 4. Discussion

Before this study, we had anticipated that *Fgf10*-heterozygous mice would exhibit intermediate phenotypes of HGs, with similar severities on both sides. In fact, as if preserving the HG on one side, there were often unilateral defects in the HG revealed by this study. Although human phenotypes of ALSG were well-documented as unilateral defects [12], and salivary gland defects were found in neonatal *Fgf10^+/−^* mice [36], mouse *Fgf10^+/−^* phenotypes in Harderian glands and regarding laterality had not been described in detail. 

This study has clarified that HG formation is retarded during development and the HG has degenerated, often unilaterally, before eyelid opening is present with reduced penetrance in the *Fgf10^+/−^* heterozygotes. The degenerated Harderian tissues are occupied by fibroblasts, melanocytes, and adipocytes possibly due to pathological changes. When heterozygous mutations reduce gene function and cause phenotypes, it is referred to as haploinsufficiency, where phenotypes are associated with variable expressivity and show reduced penetrance [37]. Although this study could not clearly show that the heterozygous HG phenotypes exhibit variable expressivity, there seems be some variation in the time course for the degeneration of the HG and the degree of tissue damage as it could be revealed by the expression of a variety of molecular markers. We had preliminary data showing that an epithelial marker, *Scd3* was expressed in the degenerating HG cells at P18. In the case of adult lacrimal glands, *Fgf10* mRNA expression becomes lower than that at the early postnatal stage, and is downregulated in chronic inflammation [38]. This suggests that the HG phenotypes in *Fgf10^+/−^* mice could vary depending on the animal age and/or the tissue regenerative capacity, such as the stem/progenitor cell number given at the developmental stage. Regarding reduced penetrance, it would be attributable to the compensatory upregulation of the unaffected *Fgf10* allele or paralogous genes that can substitute for the deleted *Fgf10* in normal individuals. In contrast to embryonic stages, qPCR analysis showed a tendency of unaltered *Fgf10* expression levels in the adult HGs from *Fgf10^+/−^* heterozygotes compared with those of wild-types (Appendix A). This result may reflect the compensatory upregulation of the intact *Fgf10* gene in adult tissue, which needs further investigation in multiple cases. 

It is crucial to know whether HGs form but later degenerate or whether they do not form at all from the beginning in certain affected cases of *Fgf10^+/−^* mice. HG primordia are absent from *Fgf10*-null embryos at E18.5 [10] (Appendix A; at E19). We examined *Fgf10-*null embryos at E15 and found that the protrusions for the future nictitating membrane (nm) are quite small (Appendix A). Since epithelial invaginations flanking the nascent nm are the early primordia for HGs, it is conceivable that HGs do not form in the complete absence of FGF10. This study does not uncover the fate of one identical HG primordium via means such as live imaging in organ cultures. Therefore, we do not conclude it decisively, but a limited number of TUNEL-positive cells were detected in the *Fgf10^+/−^* HG primordia, which is highly suggestive of the degeneration of once-formed nascent HGs. In the environment with reduced levels of FGF10, it is most likely that the proliferation of invaginating epithelial cells is insufficient, as already seen in the *Fgf10^+/−^* embryos at E15 (Figure 2D’ and Appendix A). Supportively, it was reported that the addition of FGF10 enhances and the abrogation of FGFR2b-signaling reduces cell proliferation in salivary gland explants [36]. How these reduced numbers of HG cells cannot survive and ultimately enter the elimination pathway is the next intriguing issue that needs to be addressed. 

While visualizing emerging glandular structures with the use of developing epithelial markers (*Fgfr2b*, E-cadherin) (Figure 3K,L), we found that in the nascent eyelid mesenchyme, there are a few isolated glandular cells (Figure 3H–J, arrows; Figure 3K, white arrow), which are referred to as “superficial glands”. In mammals, orbital glands located in the inner canthus include the HG, that is the “deep gland of the third eyelid” and the “superficial gland of the third eyelid” [3]. Depending on the mammalian species, the HG is absent, for example, in the okapi [39], while there have been no reports on the presence of the superficial gland of the third eyelid in mice. According to the position of the superficial gland of the third eyelid in the armadillo [40], we speculate that the primordium for the superficial gland first develops at a more superficial layer of the eyelid mesenchyme, away from the rest of the HG primordia, and disappears later in mice. 

In this study, we re-examined the expression domains of *Fgf10* along with *Fgf7* during early HG development. Relatively weak *Fgf10* expression was found abutting the superficial gland primordium and its more intense expression was found in the oval periocular domain apart from the tip of the invaginating epithelia (Figure 3A,G,K), while *Fgf7* expression was found abutting the invaginating epithelia at E15 (Figure 3I) and not at E19. These differences in expression domains might imply the distinct roles of these *Fgfs* in the HG development. *Fgf7* may have some roles in the invagination of the epithelium, while *Fgf10* may support the proliferation of the nascent superficial gland cells and the direct elongation and proliferation of the invaginated epithelia, and the HG primordia makes the FGF10 protein gradient in the periocular mesenchyme. 

We found the antisense strand expression of *Fgf10* in the intense *Fgf10* expression domains, as revealed by signals obtained when the sense *Fgf10* probe was used. It has been reported that many antisense RNA are transcribed [41], and that *Fgf10* is in fact one of them, although the mechanisms for controlling single- and sense-antisense double-strand RNA has been clarified recently [42]. 

Since transcription from a single allele is proposed to increase noise and cause stochastic delays and interruptions in transcription, due to such a non-stable and insufficient supply of *Fgf10* transcription at the developmentally critical period, HG development has likely been disrupted, nevertheless having reduced penetrance in *Fgf10^+/−^* mice. As expected, the static evaluation of mRNA expression could not capture the interrupted transcription of *Fgf10* in glandular cells lacking Harderian mesenchyme. Recent technical advancements enable the visualization of dynamic transcriptional bursting during development and the comparison of the transcriptional intensity trajectory between wild-type and *Fgf10^+/−^* fibroblasts might provide the most compelling evidence for this explanation [35].

## Figures and Tables

**Figure 2 jdb-12-00016-f002:**
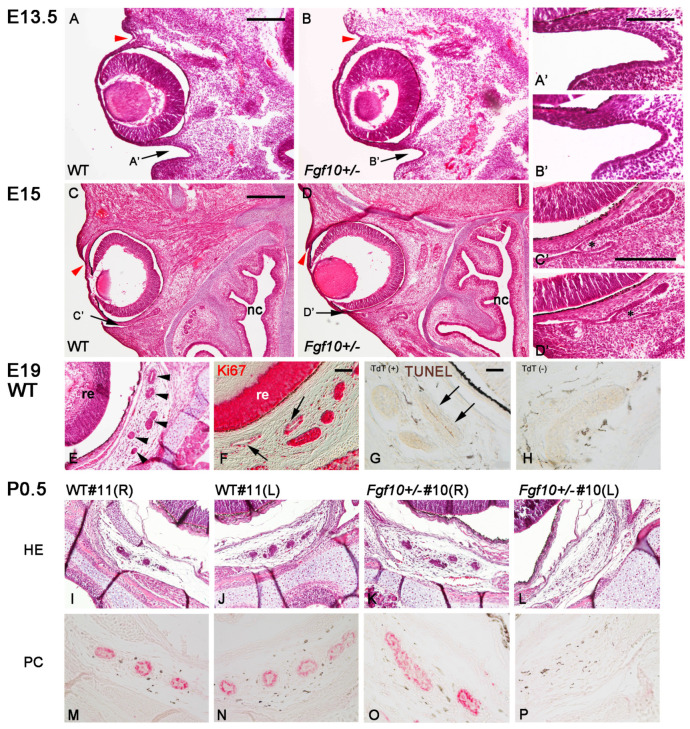
The histology of developing WT and *Fgf10^+/−^* HGs. Horizontal sections are shown. (**A**–**E**,**I**–**L**) The HE staining of the head containing the eye and the HG primordium at the indicated embryonic (E13.5, E15, E19) and P0.5 stages. The genotype of the mouse from which each section derives is indicated. The nasal cavity (nc) is shown downward. Arrows in (**A**–**D**) show the primitive conjunctival groove where the HG develops, which is enlarged in (**A’**–**D’**), respectively. Asterisks in (**C’**,**D’**) show the emerging nictitating membrane. Red arrowheads in (**A**–**D**) show where the lacrimal gland develops. (**E**) A horizontal section of the E19 WT head shows the cross-sections of the developing HGs (arrowheads), in which the upper two glandular epithelia have thin lumens. (**F**) A differential interference contrast image for the immunohistochemistry of Ki67, a proliferation cell marker, shown in red (Vector Red). Arrows show reduced Ki67 localization in the developing glands with lumens. (**G**,**H**) The TUNEL staining of the developing WT HG. A terminal deoxy-transferase (TdT)-lacking negative control is shown in (**H**). (**G**) The right elongated HG epithelium (arrows) has TUNEL-positive cells abutting the thin lumen. (**I**–**L**) The histology (HE stain) of a WT and an *Fgf10^+/−^* HG of P0.5 mice. (**M**–**P**) The immunohistochemistry of pancytokeratin (shown in red). Sections of HGs on both sides from the WT (**I**,**J**,**M**,**N**) and the *Fgf10^+/−^* (**K**,**L**,**O**,**P**) mice were used. Abbreviation: re, retina. Scale bars: 200 μm (**A**,**B**), 100 μm (**A’**–**D’**), 400 μm (**C**,**D**), 100 μm (**E**,**F**,**I**–**L**), and 50 μm (**G**,**H**,**M**–**P**).

**Figure 3 jdb-12-00016-f003:**
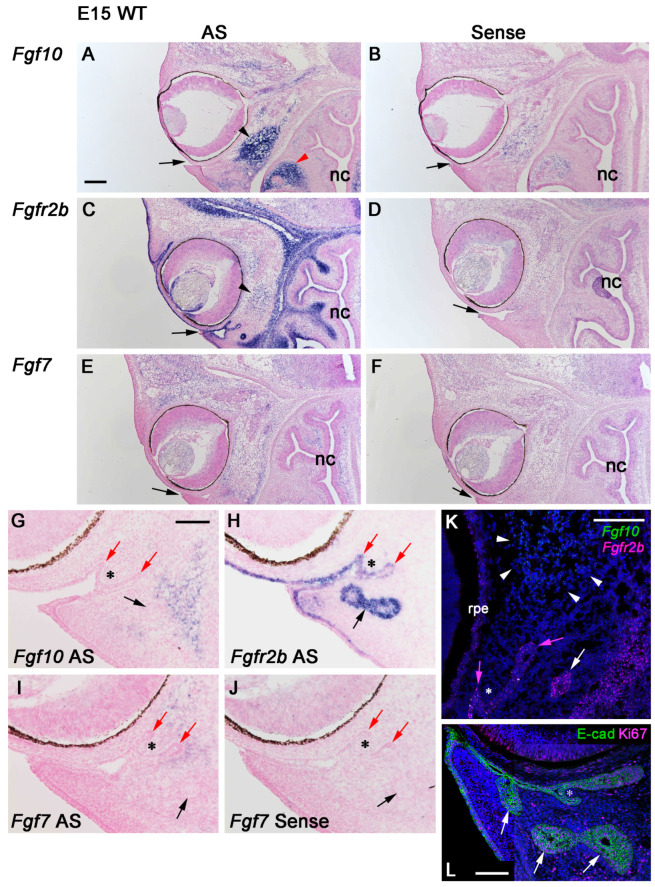
The in situ hybridization of *Fgf10, Fgfr2b,* and *Fgf7* in an E15 WT mouse head and HG primordium. A localization of mRNA is shown in dark blue. Horizontal sections are shown. (**A**–**F**) In the left panels, sections were hybridized with antisense (AS) probes for each gene, as indicated. In the right panels, sections were hybridized with sense probes (as negative controls) for each gene, as indicated. Arrows show the primitive conjunctival groove where the HG develops. The nasal cavity (nc) is shown downward. Relatively intense *Fgf10* expression in the periocular mesenchyme (black arrowhead in **A**) is enlarged in Appendix A. The red arrowhead shows *Fgf10* expression in the surrounding mesenchyme of the nasal glands. Relatively weak *Fgfr2b* expression in the periocular mesenchyme (arrowhead in **C**) is enlarged in Appendix A. (**G**–**J**) The conjunctival grooves where the HG (red arrows) and nictitating membrane (nm) (asterisks) develop. Black arrows show the superficial gland primordium of the third eyelid. Expressions of *Fgf10* (**G**), *Fgfr2b* (**H**), and *Fgf7* (**I**) are shown as indicated. In panel (**J**), the section was hybridized with a sense probe for *Fgf7*. (**K**) Dual SABER-FISH of *Fgf10* (green) and *Fgfr2b* (magenta). The *Fgf10* expression domain (arrowheads) is far from developing gland epithelia. Magenta arrows show the flanking epithelia of the nascent nm. The white arrow shows an isolated, developing superficial gland of the nm (third eyelid). (**L**) The immunofluorescence of E-cadherin (E-cad) (green) and Ki67 (magenta). Arrows show the superficial gland primordia of the third eyelid (asterisk). Cell nuclei were stained with Hoechst (**K**) or DAPI (**L**) (blue). Scale bars: 0.2 mm (**A**–**F**), 0.1 mm (**G**–**J**), 100 μm (**K**), and 50 μm (**L**).

**Figure 4 jdb-12-00016-f004:**
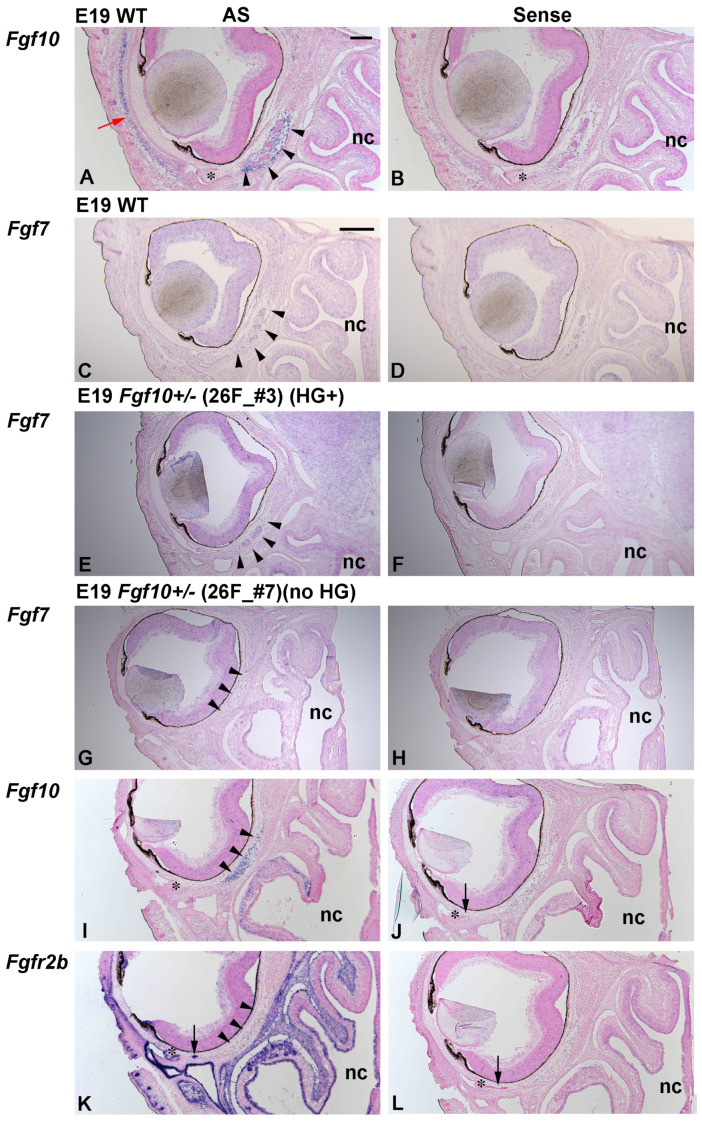
The in situ hybridization of *Fgf10, Fgf7,* and *Fgfr2b* in E19 WT and *Fgf10^+/−^* mouse HG primordia. Horizontal sections are shown. The nasal cavity (nc) is shown downward. Asterisks show the emerging nictitating membrane. Arrowheads show the Harderian mesenchyme. Left panels show sections hybridized with antisense (AS) probes for *Fgf10, Fgf7,* or *Fgfr2b,* as indicated. Arrow in (**A**) shows *Fgf10* expression in a deep mesenchymal layer of the developing eyelid. Right panels show sections hybridized with sense probes for *Fgf10*, *Fgf7,* or *Fgfr2b,* as indicated. (**A**) In the wild-type embryo at E19, *Fgf10* is expressed in the surrounding mesenchyme of the developing HGs. (**C**) *Fgf7* is not expressed in the developing Harderian tissue of the wild-type mice. (**E**) *Fgf7* expression is not upregulated in the *Fgf10^+/−^* (26F_#3 embryo) Harderian mesenchyme, where the HG primordia are developing. (**G**) *Fgf7* expression is not upregulated either in the *Fgf10^+/−^* (26F_#7 embryo) Harderian mesenchyme, where the HG primordia are missing. (**I**) In contrast, in the *Fgf10^+/−^* embryo (26F_#7) without HG primordia, *Fgf10* expression is detected. (**K**) The developing epithelium is marked by *Fgfr2b* expression. The arrow in (**J**,**K** or **L**) shows the invaginating epithelium flanking the nascent nictitating membrane. Scale bars: 0.2 mm (**A**,**B**,**I**–**L**), 0.4 mm (**C**–**H**).

## Data Availability

The data presented in this study are available on request from the corresponding author.

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
