# Peer review of "Harderian Gland Development and Degeneration in the Fgf10-Deficient Heterozygous Mouse"

_jdb, 2024, doi:10.3390/jdb12020016_

Round 1
Reviewer 1 Report
Comments and Suggestions for Authors
The study by Ikeda and colleagues characterises the presence of Harderian glands (HG) in mice haplo-insufficient for fgf10. While there has already been an established link for fgf10 for ocular gland formation, this study examines different stages of development and some markers to highlight when and where HG are present. It is a relatively brief descriptive study that extends on previous studies but lacks much novelty. It is well presented and written but there is room for improvement.
A major issue is that there is not a complete loss of HG, consistent with the fact that there is not a complete loss of fgf10. With that said, when the HG is absent, the authors claim it has degenerated, yet show no evidence of this process (no apoptosis for example).
More effort should be placed on whether HG formed in the first place. While they do examine early induction of the HG, they state it forms normally but do not show this progression. It is not clear that the early histology of invaginating surface ectoderm goes on to form HG or not. It could be that the eyes they examined all would have gone on to form HG. Authors should state how many embryos were examined at E15 to examine HG primordia formation?
If some glands are forming and others not, what could explain this? As mentioned, the authors cite early work from the Overbeek laboratory (Govindarajan et al., 2000) where the loss of fgf10 leads to an absence of HG. In fact, even earlier work from this same group showed that overexpression of fgf7 in the eye leads to ectopic gland formation, similar to HG. This is highly relevant given that fgf7 is a paralogous gene to fgf10 with both activating the same RTK (fgfr2b). Could it be that fgf7 is somehow contributing to HG formation with the reduction of fgf10?
Minor points
Some of the more relevant Supplemenatry data can be moved to the main manuscript.
Axes of graph in Figure 1M need to be labelled.
Figure 1 reference to 1N-Y in Figure legend is in bold, where all earlier letters relating to images 1A-M are not in bold.
While Figure 1 reports on the absence of HG as early as P6, authors should clearly mention the extent of serial sections through these eyes, as for Figure 2, as it is difficult to determine whether the images presented in Figure 1 are representative.
Resolution of individual Ki67 labeled cells in Figure 2F is very poor. A higher power may be more informative.
Author Response
Reply to Reviewer 1 Comments
Comment 1: A major issue is that there is not a complete loss of HG, consistent with the fact that there is not a complete loss of fgf10. With that said, when the HG is absent, the authors claim it has degenerated, yet show no evidence of this process (no apoptosis for example).
Response:
We have shown TUNEL-positive cells or nuclei in 2 cases of P6 Fgf10+/- degenerating Harderian tissues in the revised Fig. 1Z-AA’ and the source data are shown in the revised Figs. S4 and S5. Possibly because dying cells are cleared rapidly (Krieser et al., 2002, Cell Death Differ; Nagata, 2005, Annu Rev Immunol), we could only detect a few TUNEL-positive cells. This result has been described on page 4, lines 193-196 and discussed on page 12, lines 442-443.
Comment 2: More effort should be placed on whether HG formed in the first place. While they do examine early induction of the HG, they state it forms normally but do not show this progression. It is not clear that the early histology of invaginating surface ectoderm goes on to form HG or not. It could be that the eyes they examined all would have gone on to form HG. Authors should state how many embryos were examined at E15 to examine HG primordia formation?
Response:
We reexamined the early histology of invaginating surface ectoderm at E15. The number of examined embryos is: 3 wild-type, 3 Fgf10+/-, and 2 Fgf10-/-. Careful observation of serial sections has revealed that the wild-type tip of the epithelium is expanded with multilayered cuboidal/columnar cells on both sides in all three embryos. In contrast, the Fgf10+/- invaginating epithelium appeared less expanded and exhibited delayed development, although the nictitating membrane (nm) was emerging on both sides in all three embryos. The two Fgf10-/- embryos examined exhibited only small protrusion corresponding to the nascent nm. All these data have been shown in the revised Fig. S6. In Fig. 2, panels C-D’ have been replaced by better ones. This result has been described on page 6, lines 245-248.
Comment 3: If some glands are forming and others not, what could explain this? As mentioned, the authors cite early work from the Overbeek laboratory (Govindarajan et al., 2000) where the loss of fgf10 leads to an absence of HG. In fact, even earlier work from this same group showed that overexpression of fgf7 in the eye leads to ectopic gland formation, similar to HG. This is highly relevant given that fgf7 is a paralogous gene to fgf10 with both activating the same RTK (fgfr2b). Could it be that fgf7 is somehow contributing to HG formation with the reduction of fgf10?
Response:
Regarding the reviewer’s questions, we performed qPCR and in situ hybridization (ISH). qPCR results (Fig. S10B, C) showed expression of Fgf7 was not upregulated in the Fgf10+/- embryonic tissues at E19 and at E15. ISH showed there was no alteration in Fgf7 expression between wild-type and Fgf10+/- heterozygotes. In adult HGs (Fig. S10D), the expression of Fgf10 and Fgf7 was variable in the heterozygotes, suggesting that environmental cues after birth might influence the expression of these Fgf genes, which we would like to investigate in the future study. We have cited Lovicu FJ et al., Mech Dev.1999 Oct;88(1):43-53 as reference #33, the above result has been described on page 11, lines 375-385, and discussed on page 12, lines 429-433.
Minor points
Comment 4: Some of the more relevant Supplementary data can be moved to the main manuscript. Axes of graph in Figure 1M need to be labelled.
Response:
We have replaced Fig. 1H-J with Supplementary Figure, Fig. S1A-C. The y axis of Figure 1M has been labeled.
Comment 5: Figure 1 reference to 1N-Y in Figure legend is in bold, where all earlier letters relating to images 1A-M are not in bold.
Response:
We have revised the legend (made the letters (A-M) in bold) accordingly (page 5, Figure 1 legend).
Comment 6: While Figure 1 reports on the absence of HG as early as P6, authors should clearly mention the extent of serial sections through these eyes, as for Figure 2, as it is difficult to determine whether the images presented in Figure 1 are representative.
Response:
We have added the following description to the legend of Fig. S1V-Y:
“Consecutive sections through upper to lower horizontal planes were observed and representative sections are shown. “ on page 6, lines 217-218.
Comment 7: Resolution of individual Ki67 labeled cells in Figure 2F is very poor. A higher power may be more informative.
Response:
Figure 2F has been replaced by a DIC image to show Ki67 localization more clearly. Ki67 was visualized by immunohistochemistry with Vector Red having a red fluorescence. Epifluorescence images were also shown in Fig. S7A-C to show the presence of DAPI-stained cell nuclei without Ki67 localization.
Reviewer 2 Report
Comments and Suggestions for Authors
In this manuscript, Ikeda and colleagues pinpoint key aspects of the Harderian gland atrophy in Fgf heterozygous mouse model. For that, the authors used a series of histology techniques, and ultimately showed that Fgf10 delay Harderian gland development in early neonatal period. That said, I would like to bring the author’s attention to the following:
Major comments
1. Based on the Supplementary Table 3 showing all phenotypes observed in the Fgf10 +/- group, I would strongly suggest the authors to include a quantitative RT-PCR analysis showing the mRNA expression reduction observed in the heterozygous group compared to the WT and increase compared to the KO. Or at least a western blot to assess the protein levels in each group.
2. Recently, Finburgh and colleagues (Finburgh et al., 2023- IOVS) have shown that FGF10 downregulation is somehow linked to chronic inflammation in eye tissues. It was also shown that Fgf10 mRNA expression decreases over time. In the present manuscript it is shown increased cell death in the heterozygous mice group compared to the WT group (TUNEL In Fig 1 P/Q and Ki67Fig 2 F). I suggest the authors better elucidate possible mechanisms involved in the increased cell death and, how Fgf10 mRNA expression over time, could play a role in that case.
3. There is a noticeable global eye phenotype in the Fgf10 +/- mice, as seen in Figure 1H showing ocular lens and eye cup compromised morphology (FGF signaling in lens development review: Robinson, 2006). The authors do not mention throughout the manuscript other structures also affected in Fgf10+/- mouse model. Are those morphological features also seen unilaterally? That additional information could potentially help to better elucidate the phenotype seen.
4. The fact that the Fgf10 heterozygous mice display reduced penetrance in the observed phenotype indicates that other endogenous FGF signaling pathways could play an important role, as seen in the salivary gland (Jaskoll et al., 2005). I wonder why the authors concluded that it could be due to a transcriptional bursting instead.
Minor comments
1. Abstract: remove references, or check journal guidelines
2. I would suggest the authors include a scheme figure showing the Fgf10 +/- mouse exon disruption (WT, knockout and het) and how it compares to the previous study (Line #76, Puk et al, 2009) deletion.
3. Figure 1 seems to be misplaced in the text.
Lines #192- 193: I would suggest including the picture of the slit-eye phenotype observed in the Fgf10+/- mice.
Author Response
Reply to Reviewer 2 Comments
Comment 1: Based on the Supplementary Table 3 showing all phenotypes observed in the Fgf10 +/- group, I would strongly suggest the authors to include a quantitative RT-PCR analysis showing the mRNA expression reduction observed in the heterozygous group compared to the WT and increase compared to the KO. Or at least a western blot to assess the protein levels in each group.
Response:
We performed qPCR to examine the relative expression level of Fgf10 in the Fgf10+/- mouse (n = 3) compared with wild-type (n = 2) and Fgf10-null (n = 3) embryos at E15 (Supplementary Fig. S10C). Fgf10 mRNA expression was reduced in the Fgf10+/- mice compared with wild-type, while it was not detected in the Fgf10-null embryos. In the case of E19 heads, Fgf10 expression was significantly reduced in the Fgf10+/- heads (n = 6), compared with wild-type (n = 4). (Fig. S10A) In contrast, using 19-week Harderian glands (HGs), relative expression levels of Fgf10 were variable between the wild-type and Fgf10+/- tissues, in which one degenerated HG had undetectable expression level of Fgf10 or Gapdh (Fig. S10D). These data suggest the expression level of Fgf10 is reduced in the embryonic Fgf10+/-tissues, while it seems to be variable in adult wild-type and Fgf10+/- normal-looking Harderian tissues. This result has been described on page 11, lines 378-381 and discussed on page 12, lines 429-433.
Comment 2: Recently, Finburgh and colleagues (Finburgh et al., 2023- IOVS) have shown that FGF10 downregulation is somehow linked to chronic inflammation in eye tissues. It was also shown that Fgf10 mRNA expression decreases over time. In the present manuscript it is shown increased cell death in the heterozygous mice group compared to the WT group (TUNEL In Fig 1 P/Q and Ki67Fig 2 F). I suggest the authors better elucidate possible mechanisms involved in the increased cell death and, how Fgf10 mRNA expression over time, could play a role in that case.
Response:
Only Harderian gland tissue we had was at 19 weeks and using this, we performed qPCR and tried to compare the Fgf10 expression level between different genotypes as mentioned above. In fact, we could not obtain other stage, comparable Harderian glands for qPCR during this revision time, and therefore we would like to challenge the suggested issue in the next study. We have added discussion on FGF10 downregulation and inflammation in the lacrimal gland (Finburgh et al., 2023- IOVS) (page 12, lines 422-427). When it comes to the Fgf10 expression at the time of HG formation before birth, such as at E15 and E19, the expression domain is restricted to a very tiny potion, which we call Harderian mesenchyme bordered by a capsule- or a sheath-like tissue. Since in Fgf10+/- mice, HGs degenerate unilaterally unexpectedly early, by P6, and their development retardates by E19, we think the expression of Fgf10 before birth is critical. To obtain quantitative data for Fgf10 expression in these tiny portions, the best way would be to laser-microdissect the Harderian mesenchyme and use the dissected sections for qPCR analysis, which we would like to perform in the next study.
Comment 3: There is a noticeable global eye phenotype in the Fgf10 +/- mice, as seen in Figure 1H showing ocular lens and eye cup compromised morphology (FGF signaling in lens development review: Robinson, 2006). The authors do not mention throughout the manuscript other structures also affected in Fgf10+/- mouse model. Are those morphological features also seen unilaterally? That additional information could potentially help to better elucidate the phenotype seen.
Response:
According to the previous reports, conditional deletion of Fgfr2b at the lens placode stage exhibits defects in lens formation such as apoptosis and fiber cell degeneration (Li C et al., Dev Dyn 2001; Robinson ML, Semin Cell Dev Biol, 2006). Our data show the distinct Fgfr2b expression in the lens epithelial to fiber cells at E15 (Fig.3C), supporting the involvement of FGFR2b signaling in lens development. In contrast, Fgf10 or Fgf7 mRNA was not detected in the vicinity of lens cells at this stage, although it has been thought FGF proteins in the aqueous humor and in the vitreous humor regulate lens epithelial cell proliferation and fiber differentiation (McAvoy et al., Exp Eye Res, 2017). Since the reviewer pointed out a possible lens phenotype (Fig. 1H: Fgf10+/- #5 mouse, right eye in the previous version, which has been transferred to revised Supplementary Fig. S1A-C), we observed histology of lens fiber cells in 74-week Fgf10+/- eyes, compared with wild-type. Although it might reflect aging and/or due to artifacts raised in the histological preparations, there seemed to be elongation defects in certain cases as summarized in a pdf file for reviewers only. We think that it requires detailed assessment with better histological preparations and would like to make it a next important issue.
Comment 4: The fact that the Fgf10 heterozygous mice display reduced penetrance in the observed phenotype indicates that other endogenous FGF signaling pathways could play an important role, as seen in the salivary gland (Jaskoll et al., 2005). I wonder why the authors concluded that it could be due to a transcriptional bursting instead.
Response:
We examined the relative expression level of Fgf7, encoding a most related FGF to FGF10, in the Fgf10+/- and Fgf10-null mice by qPCR to know whether the expression of Fgf7 was upregulated to compensate the absence of Fgf10 expression. Figure S10B shows that Fgf7 was not significantly upregulated in the Fgf10+/- heads at E19. Figure S10C shows the expression level of Fgf7 appeared variable but not upregulated even in the Fgf10-null embryos at E15. We also performed in situ hybridization to see whether Fgf7 mRNA became detected in the Harderian mesenchyme (HG+ and HG- samples) in the Fgf10+/- and Fgf10-null (HG-) heads at E19 (Figure 4C-H; Supplementary Fig. S11), but the expression pattern of Fgf7 was not changed in the heterozygote or Fgf10-null heads, compared with wild-type. Based on these data, we would like to refer to a transcriptional bursting as a possible mechanism for reduced penetrance in the observed phenotype, but according to the reviewer’s suggestion, we have firstly referred to the possibility of compensation by other Fgfs as Fgf7. This has been described on page11, lines 375-385.
Minor comments
Comment 1: Abstract: remove references, or check journal guidelines.
Response:
References have been removed from Abstract.
Comment 2: I would suggest the authors include a scheme figure showing the Fgf10 +/- mouse exon disruption (WT, knockout and het) and how it compares to the previous study (Line #76, Puk et al, 2009) deletion.
Response:
We have included the scheme figure in the revised Supplementary Fig. S2A.
Comment 3: Figure 1 seems to be misplaced in the text.
Response:
Figure 1 has been placed after section 3.1.
Comment 4: Lines #192- 193: I would suggest including the picture of the slit-eye phenotype observed in the Fgf10+/- mice.
Response:
We have added the photos showing the slit-eye phenotype in Fig. S2B.
Round 2
Reviewer 1 Report
Comments and Suggestions for Authors
The authors had considered and acted on the constructive comments suggested by the review. They have reviewed data, added new data nad incorporated it effectively, addressing this in the Discussion as well. In light of this, the manuscript is much improved and will be a welcome addition to the role of fgf in ocular gland development.
Reviewer 2 Report
Comments and Suggestions for Authors
I thank the authors for addressing my previous comments. I believe the manuscript is suitable for publication.
Comments on the Quality of English LanguageQuality of English is good.